# Making the Argument for Intact Cord Resuscitation: A Case Report and Discussion

**DOI:** 10.3390/children9040517

**Published:** 2022-04-06

**Authors:** Judith Mercer, Debra Erickson-Owens, Heike Rabe, Karen Jefferson, Ola Andersson

**Affiliations:** 1Neonatal Research Institute, Sharp Mary Birch Hospital for Women and Newborns, San Diego, CA 92123, USA; 2College of Nursing, University of Rhode Island, Kingston, RI 02881, USA; debeo@uri.edu; 3Brighton and Sussex Medical School, University of Sussex, Brighton BN2 5BE, UK; heike.rabe@nhs.net; 4American College of Nurse-Midwives, Silver Spring, MD 20910, USA; kjefferson@acnm.org; 5Department of Clinical Sciences Lund, Paediatrics, Lund University, 221 85 Lund, Sweden; ola.andersson@med.lu.se

**Keywords:** intact cord resuscitation, placental transfusion, cord clamping, perfusion, vagus nerve

## Abstract

We use a case of intact cord resuscitation to argue for the beneficial effects of an enhanced blood volume from placental transfusion for newborns needing resuscitation. We propose that intact cord resuscitation supports the process of physiologic neonatal transition, especially for many of those newborns appearing moribund. Transfer of the residual blood in the placenta provides the neonate with valuable access to otherwise lost blood volume while changing from placental respiration to breathing air. Our hypothesis is that the enhanced blood flow from placental transfusion initiates mechanical and chemical forces that directly, and indirectly through the vagus nerve, cause vasodilatation in the lung. Pulmonary vascular resistance is thereby reduced and facilitates the important increased entry of blood into the alveolar capillaries before breathing commences. In the presented case, enhanced perfusion to the brain by way of an intact cord likely led to regained consciousness, initiation of breathing, and return of tone and reflexes minutes after birth. Paramount to our hypothesis is the importance of keeping the umbilical cord circulation intact during the first several minutes of life to accommodate physiologic neonatal transition for all newborns and especially for those most compromised infants.

## 1. Introduction

Around the world, 2.9 million babies die each year at birth from perinatal asphyxia, and a third die within the first day [1]. In many cases, the newborns received immediate or early cord clamping, a non-evidence-based intervention [2]. Waiting to clamp the umbilical cord at birth results in a 30% decreased incidence of death for preterm infants [3,4,5,6]. Enhanced blood volume from placental transfusion may be the most important factor that prevents death in these infants [7]. Immediate and early cord clamping can prevent the transfer of essential fetal–neonatal blood volume and interfere with its associated benefits.

Performing resuscitation with an intact cord provides a continuous connection to the placenta, allowing the depressed newborn to have ongoing circulation of fetal–neonatal blood. This action provides cardiovascular stabilization, exchange of oxygen and carbon dioxide, and other support during resuscitation [8,9]. The amount of blood available is significant: for a term newborn, it is about 30% of the fetal–placental blood volume [10]; for a preterm infant, it is up to 50% [11]. Loss of this large blood volume can result in severe hypovolemia and ischemia as the newborn adapts to life without aid from the placenta [12]. Intact umbilical cord milking supports the transfer of some of the blood volume to the newborn quickly, which is beneficial but does not provide for continued connection or respiration [13]. Two recent papers on intact cord resuscitation have reported improvements in infants and newborn animals needing resuscitation. Andersson et al. found higher Apgar scores at 5 and 10 minutes after a three-minute wait for newborns who needed resuscitation compared to those receiving immediate cord clamping [14] and improved neurodevelopment after two years [15]. Polglase reported that in post-asphyxial newborn lambs, a 10-minute delay in cord clamping (after return of systemic circulation) prevented an overshoot of blood pressure (post-asphyxial rebound hypertension). This overshoot was not prevented by either a one-minute wait or immediate cord clamping [16].

Resuscitation with an intact cord is fairly routine in out-of-hospital settings in the US and Canada, but few in-hospital providers have attended such births [17,18]. However, historic references of intact cord resuscitation exist as far back as Aristotle [19]. For those with experience-derived knowledge of intact cord resuscitation, it seems illogical to cut the cord immediately and remove nonbreathing newborns from their only source of respiratory and circulatory support during this critical time [17,18].

We present a case of a moribund newborn who was successfully resuscitated with an intact cord and then discuss the probable physiology underlying this uncommon event. (see Figure 1) [2].

## 2. Case Presentation

A 28-year-old woman having her second infant was in labor at term and progressed rapidly to full dilatation with intact membranes. During second stage, the baby’s head remained high between contractions and recoiled after descending to the +3 station with each push. After artificial rupture of the membranes, the infant descended, crowned, and delivered in one contraction. A tight double nuchal cord was unwound immediately, and the cord was kept intact. The newborn was extremely pale white with no tone, reflexes, or respiratory effort and was placed on the bed between the mother’s legs, where the provider bulb suctioned, dried, and stimulated the infant. A check of the heart rate at about 20 s revealed that it was over 100, the cord appeared full and pulsatile, and the infant was visibly gaining color. The provider continued to stimulate the baby with no response. At about 2 minutes (exact time was not recorded), the baby, now appearing well perfused, moved his left arm upon his chest, opened his eyes, took an easy breath, and quickly flexed his other extremities. The baby never vocalized a cry, even with strong stimulation, but continued gentle, even, and shallow respirations with clear lungs. The cord was clamped and cut about 20 minutes after birth, and the infant had a normal newborn course. At one year, the mother reported no concerns with the infant. (Adapted from case first published in Downe in 2008 [17]).

## 3. Discussion

Adequate circulation, as well as breathing, is essential for a successful neonatal transition. We are proposing the following points, which are illustrated in Figure 2. First, we discuss lung readiness in preparation for the initial breath and the importance of circulation as well as breathing for transition, and we describe how a placental transfusion may return blood volume to provide enhanced perfusion for all organs. Next, we provide evidence supporting our hypothesis that enhanced perfusion may stimulate the vagus nerve to cause vasodilation in the lung, thereby reducing pulmonary vascular resistance and increasing blood flow before the first breath. Last, we suggest that enhanced perfusion to the presented case infant’s brain enabled breathing and establishment of consciousness, tone, and reflexes. Remaining attached to the umbilical cord after birth provided the infant full access to his fetal–placental blood volume to accomplish transition. Each point is described, explained, and referenced in the following text.

### 3.1. Lung Readiness and Preparation for the First Breath

#### 3.1.1. Lung Growth and Development in the Fetus

Lung growth in fetal life depends on the alveoli being highly distended with a large volume (20–30 mL/kg) of fetal lung liquid [20,21], which creates lung expansion along with mechanical stress/stretch that stimulates and promotes normal growth and development. If not for this liquid, the fetus would develop hypoplastic lungs [20]. The lungs maintain a low pH (6.27) in the alveolar fetal lung liquid constantly throughout fetal life [22]. It is important to clarify that fetal lung liquid is not amniotic fluid (Table 1) [20].

Fetal lung liquid is generated and maintained by the alveolar epithelial cells. These cells secrete chloride ions into the alveolar breathing spaces filled in utero with fetal lung liquid [22]. In most vascular beds, acidosis causes vasodilation. However, in the pulmonary circulation, acidosis evokes vasoconstriction supporting vascular resistance [23]. At birth, a switch must occur so that the alveolar epithelial cells stop secreting chloride into the fetal lung liquid and immediately begin to pump fetal lung liquid into the interstitium by means of sodium channels.

#### 3.1.2. Anatomy of the Respiratory Tract and Blood Supply

The process of fetal lung liquid removal from the interstitium is informed by the anatomy of the alveolus and its blood supply [20]. About 80% of an adult’s alveoli develop after birth. However, in the term newborn, around 50 million alveoli are covered by approximately 47 billion capillary segments [20] or a ratio of 1 alveolus to around 900 capillary segments. The capillaries are so dense that an almost continuous sheet of blood encircles each alveolus, allowing for maximal gas exchange. There is only a very thin (0.5 μm) interstitial space between an alveolus and the surrounding capillary network which facilitates gas exchange.

The large number of capillaries segments covering the alveoli are cemented to the alveoli by an extracellular matrix [24]. As the dense alveolar capillary network fills with blood, the capillary plexuses expand with blood and become erect [7,25,26]. This process opens the alveoli and provides the “scaffolding” structure that maintains alveolar expansion and likely prevents the alveoli from closing or collapsing on expiration [25,26].

The alveolar capillaries adapt to the sudden influx of blood at birth by increasing the diameter of the lumen, but they do not change the diameter of the capillary itself (Figure 3) [27,28]. This allows the overall lung volume to remain the same. Figure 3 shows the immediate decrease in wall thickness of the porcine pulmonary inter-acinar arteries as the lumens dilate with blood. This allows transfer and inflow of the blood that formerly circulated through the placenta for respiration during fetal life.

#### 3.1.3. Switching from Fetal to Neonatal Respiratory Function

At birth, the alveolar membrane makes an instantaneous switch from producing and maintaining the fetal lung liquid to excreting it to the interstitium [28]. We suggest that the mechanical stress/stretch from the reallotment of the residual placental blood volume into the alveolar capillary network plays a major role in the switch [22]. Perks et al., studying fetal goats at different gestational ages, found that lung expansion from the infusion of saline reduced production and reabsorption of fetal lung liquid [29]. They suggested that expansion of the lung activates the ion pump, Na^+^/K^+^-ATPase, which generates a transepithelial osmotic gradient that causes the movement of fluid out of the alveolar airspace to the interstitium [29]. It is likely that the enhanced vascular perfusion provides mechanical stress/stretch through vasodilation in the alveolar capillary network. This continues the distension, vital in fetal life, to support progressive neonatal lung development and growth [30]. Effects of the essential stretch over time on the vessels surrounding the alveoli can be seen in the porcine lung over the first five minutes (Figure 3) and hours (Figure 4) following normal birth [27,31].

After the switch, both Type I and II alveolar epithelial cells pump sodium ions out of the alveolar spaces into the interstitium, generating the driving force for removal of the fetal lung liquid from the alveoli rapidly after birth [22]. However, this fetal lung liquid cannot stay in the interstitium without compromising air exchange and requires rapid removal. The pulmonary circulation (alveolar capillary network) absorbs most of the residual fetal lung liquid from the interstitium after birth [22]. Thus, there are two rapid components to the process of fetal lung liquid removal: (1) transepithelial flow of the fetal lung liquid into the interstitium and (2) removal of the fetal lung liquid from the interstitium by the alveolar capillaries and into the systematic blood stream through the alveoli capillary network [22]. Evidence for the validity of the rapid removal of acidic fetal lung liquid is suggested by Wiberg’s finding of a slight decrease in pH that has been reported in newborns with continued cord circulation [32,33]. The decrease in pH was significant in blood from the intact umbilical arteries at 45 s of life but not until after 90 s in the umbilical vein. This indicates that the newborn was the source of the change in pH and not the placenta. It was similar in infants born either vaginally or by cesarean section and occurred despite good oxygenation [32]. These findings show that the acidic fetal lung liquid must enter the infant’s alveoli–capillary network blood supply rapidly. The change in pH is statistically, but not clinically, significant. However, for an infant with immediate or early cord clamping, the placental blood flow will not be available to assist with removal or dilution of the acidic lung liquid.

#### 3.1.4. Breathing after Birth

The first breath our case newborn experienced does not fit the prevailing paradigm, which states that high initial negative pressure is required for lung opening to decrease the pulmonary vascular resistance at birth and push the lung fluid out of the alveolar spaces [34]. It is assumed that hypoxia will develop in the non-breathing infant, but in our case newborn, placental respiration overcame this as reflected in the newborn’s heart rate and improving color. This is similar to the reliance of placental respiration and umbilical circulation during procedures using the ex-utero intrapartum technique (EXIT) [35]. The EXIT procedure consists of hysterotomy and partial delivery of the head and upper torso of the fetus while maintaining uteroplacental gas exchange from the umbilical cord. Normally, the procedure is said to be possible for 60 minutes, but cases with sufficient placental respiration and umbilical circulation for up to 150 minutes have been reported [36].

Recent studies report that applying a face mask or even nasal prongs to a newborn immediately after birth may inhibit initial breathing and reduce the heart rate [37,38,39]. Applying these devices induces vagally mediated facial reflexes that inhibit spontaneous breathing. The trigeminocardiac reflex and the laryngeal chemoreflex can also be elicited by air flow, leading to glottal closure [40]. Most likely, the mammalian diving reflex is triggered; it consists of breathing cessation (apnea), a dramatic slowing of the heart rate (bradycardia), and an increase in peripheral vasoconstriction. The diving reflex is thought to conserve vital oxygen stores and thus maintain life by directing perfusion to the two organs most essential for life: the heart and the brain [41].

A puzzling finding, given the current paradigm, is the report of reduced blood flow to the heart with breathing movements. A recent newborn lamb experiment found that the umbilical venous flow was markedly reduced with each breath [42]. In another newborn lamb study, a 30-s sustained inflation, even with delayed cord clamping, prevented the lamb’s blood from flowing into the lung via the pulmonary artery and inferior vena cava [43]. A randomized control trial comparing sustained inflation with usual ventilation in human preterm infants was stopped because more early deaths occurred in the infants with sustained inflation [44].

These findings demonstrate that early or forced breaths may interfere with a physiologic transition by blocking the essential increased blood flow to the heart. Aerating the lung is critical for the success of neonatal transition but doing so before the lung has been adequately perfusion may be harmful. A heart rate above 100 or increasing indicates wellbeing and, as in our case, continued placental respiration [17]. Thus, attempting to start ventilation immediately after birth may compromise initial and early breathing as well as transition [45]. It is much easier to push air into lungs when the alveolar capillary circulation is fluid-filled [25,46].

Awareness of the heart rate allows a clinician to assess whether one can wait longer than one minute before assisting with ventilation. Waiting longer may be especially beneficial when there has been some cord compression as with a nuchal cord, shoulder dystocia, or occult cord. While some say that the main cause of persistent neonatal bradycardia is inadequate ventilation, we suggest that the loss of the available blood volume from immediate or early cord clamping is a likely cause in many cases. Bhatt has clearly demonstrated the cost of immediate cord clamping to the newborn by showing a 50% reduction in right ventricular output, heart rate, and pulmonary blood flow immediately after birth in preterm lambs [8]. Both Katheria and Nevill have reported that for preterm infants, there is no harm in waiting 60 s after birth to deliver the first assisted breath [2,47,48]. They found that 90% of the infants had breathed on their own by 60 s and that no clinical benefit was reported from assisted breathing before that time.

### 3.2. Circulation

#### 3.2.1. Contents of Cord Blood

Maintenance of an intact cord allowed our case newborn to receive his full allotment of blood volume. This warm, oxygenated residual placenta blood amounts to about 15–20 mL/kg of red blood cells - enough to provide the term infant additional oxygen-carrying capacity and adequate iron for four to 12 months [49,50]. The blood also contains up to a billion of several kinds of stem cells providing an autologous transplant, which may reduce the neonate’s susceptibility to both neonatal and age-related diseases [51]. Neuroprotective progesterone in the term neonate’s blood at birth is almost two-times higher than the mother’s level. It likely causes vasodilation, which can help to distribute the large amount of placental transfusion throughout the body [7,52,53]. In addition, this blood, unique in its composition for the newborn, contains many essential factors such as cytokines, growth factors, and important messengers that support and drive the process of transition (see Table 2) [11,54]. The enhanced perfusion provides higher pulmonary artery pressure [55] and mechanical stimuli, which causes electrochemical signaling that is likely essential for the normal function, maturation, and maintenance of all organs [7,30].

#### 3.2.2. Placental Transfusion Enables the Return of Blood Volume

As the head is born and uterine fibers shorten, the uterus contracts with more pressure around the placenta. The strong expulsive contractions at the end of second stage or early third stage begin to force placental blood to the infant [56]. In humans, the intervals between these contractions allows for continued placental gas and nutrient exchange for the fetus/newborn [57,58]. Umbilical circulation continues to flow through the intact umbilical arteries for several minutes [59]. Boere measured the blood flow with doppler ultrasound in the intact umbilical cord of healthy term babies. Before cord clamping, over 80% of the infants in the study had umbilical arterial flow for a mean of four minutes, while almost half (43%) still had arterial flow when the cord was clamped at six minutes [59]. This information is not considered in current resuscitation protocols for infants of all gestational ages that imply immediate or early cord clamping and the care of the newborn away from the mother’s bedside [2]. For our case infant, maintaining an intact cord was life-sustaining and likely avoided morbidity and possible mortality (Figure 1).

#### 3.2.3. Effect of Occlusion of the Umbilical Cord

Nuchal cord compression likely caused much of our case infant’s blood volume to be sequestered in the placenta, resulting in the appearance of extreme paleness and unconsciousness [60]. Unwrapping the cord and leaving the cord intact allowed the blood that was sequestered in the placenta to reperfuse the infant’s body so that he did not lose the essential blood volume necessary to oxygenate his brain and other vital organs. Although the cord felt pulseless initially, it began to pulse within 15 to 20 s, and the infant’s body began to regain color.

When an umbilical vein is occluded, there is a possibility of a net transfer of blood from the fetus to the placenta (see Figure 5) [60]. The soft-walled vein can be compressed easily, preventing the return of blood volume to the fetus, while the more muscular-walled higher pressure arteries allow for a maintained flow from the fetus to the placenta. If severe enough, the loss of blood volume in the body can cause the fetus/newborn to appear to be unconsciousness and will result in extreme paleness at birth due to hypovolemia. Severe hypovolemia may lead to a sudden unexpected neonatal asystole, especially if the cord is severed immediately [60,61,62]. The neonatal status may be compromised by the uncorrected physiologic effects of blood volume loss leading to hypoxia.

### 3.3. Full Perfusion, the Vagus Nerve, and Pulmonary Vascular Resistance

One of the key features of the normal fetal-to-neonatal transition at birth is a reduction in the pulmonary vascular resistance to allow an increase in pulmonary blood flow. Credit has been given to lung aeration for decreasing the pulmonary vascular resistance at birth, although the mechanisms causing this have been debated for decades [63]. Surprisingly, Lang et al. found that severing the vagal nerve prevented the previously observed increase in pulmonary blood flow with partial lung aeration [64]. Compared to control newborn rabbits, he found that vagotomized rabbit newborns had little or no increase in pulmonary blood flow when ventilated with air or nitrogen gas. Using 100% oxygen for ventilation only partially mitigated the effect of the vagotomy. This new information suggests that the initial dramatic fall in pulmonary vascular resistance likely does not occur with ventilation/breathing alone but that vagal stimulation must play a significant role. Lang et al. suggests that given the importance of lowering the pulmonary vascular resistance, which is so essential to initiate blood flow to the lung, there are likely multiple overlapping mechanisms to ensure that this transition happens [64].

#### 3.3.1. The Vagal Stimulation Hypothesis

The newborn illustrated in our case did not have any assisted breathing, and his first breaths were easy and quiet. The most plausible hypothesis for this is that enhanced perfusion from the placental transfusion stimulates the volume-sensing cardiopulmonary mechanisms [65]. As more blood volume enters the umbilical vein and moves through the inferior vena cava and into the right atrium, it signals the sinoatrial (SA) node (and other receptors) that blood flow is enhanced. The vagus nerve is stimulated via stretch mechanisms (baroreflex) to effect vasodilation in the lung, opening the pulmonary arteries and thus decreasing pulmonary vascular resistance [66]. This would have allowed blood flow into the alveolar capillary network supporting capillary erection, lung fluid removal, and likely an effortless first breath.

#### 3.3.2. The Autonomic Nervous System

The autonomic nervous system is composed of two parts: the parasympathetic and sympathetic systems. Although their effects are opposed to each other, they work reciprocally to bring about the necessary responses to internal and external stimuli, continuously balancing each other to maintain homeostasis [67]. The vagal sensory input (part of the parasympathetic system) can detect visceral and environmental features and can adjust the central regulation of autonomic function to restore homeostasis. It can dampen the sympathetic activation to protect the oxygen-dependent central nervous system from the metabolically conservative defensive reactions (stress response, vasoconstriction, flight or fight, fainting).

In our featured case, decreased blood volume due to sequestration to the placenta likely reduced brain stem perfusion, thereby inhibiting consciousness, tone, and appropriate reflexes. When the compression of the cord was released immediately after birth and left intact, it allowed the placental residual blood to reallocate into the baby’s body as validated by the robust pulsating cord. The returning blood volume distended the right heart’s mechanical receptors, likely causing activation of the vagal complex to dilate the pulmonary arteries and the alveolar capillary network.

This idea is counter to the prevailing paradigm that it is the sympathetic nervous system that plays the predominant role in neonatal transition [68]. We suggest that soon after birth, the vagus nerve must dominate, as visceral homeostasis is incompatible with sympathetic domination [69]. It is important that the infant is not in a “fight or flight” response mode to initiate some of the first tasks after a successful transition (sucking and bonding). In the face of severe compromise, the fetus maintains autonomic nervous system function, which can be balanced with good perfusion [58].

### 3.4. Effects of Enhanced Perfusion on Consciousness, Breathing, Tone, and Reflexes

Impaired perfusion of the brainstem results in syncope, unconsciousness, decreased muscle tone, and lack of reflexes [70]. For our case newborn, the return of good perfusion to the heart and brain, sensed by various receptors in the body, resulted in domination of the vagal reflex, which reversed the negative changes of impaired perfusion. The return of tone and reflexes with breathing supports the idea that good perfusion of the brain is essential in an infant who appears unconscious and toneless at birth.

Our case newborn’s one-minute Apgar score was two for a heart rate over 100. By two minutes of life, he opened his eyes and started easily breathing without assistance at the same time. The tone returned over his entire body, and he had good color and reflexes, resulting in a five-minute Apgar of 10. The case infant started breathing on his own without ventilation, gasping, or crying after his body was perfused. It is likely that some newborns receive at least some of their placental transfusion before they breathe for the first time, and this was clearly illustrated in our case [17,18]. The fetal–placental blood volume to the infant, from sustained cord circulation, supplies a higher level of systemic oxygen that stimulates the respiratory centers of the brain [68].

### 3.5. One Caveat: A Note of Caution

For an infant with an acute sequestration of blood in the placenta from cord compression, nuchal cord, or shoulder dystocia, the benefit from the return of this blood volume to the infant, via placental transfusion, is very important. However, there are a few infants for whom allowing for continued cord circulation may not make a difference. For those newborns, who may have had subtle chronic distress going on for hours during labor, the blood volume loss (mainly plasma) to the placenta may be significant. This can occur because of the fetal stress response during a difficult labor [58]. When a fetus is stressed, the blood pressure becomes elevated, and more of the fetus’s blood reallocates to the placenta [71]. A transplacental water flux causes an increase in water loss from the fetus to the mother, reducing the fetus’s total blood volume over time. Hypertonicity of contractions may be an important issue here as the fetus needs at least 1 or more minutes in between contractions to recover from the effects of the contractions and to prevent hypoxemia [58]. In a near-term sheep model, Brace reported a 7% blood volume loss from the fetus to the placenta in only 30 min by reducing the oxygen content for the near-term pregnant sheep. After returning the pregnant sheep to room air, it took about 30 min for the fetal blood volume to return to normal. However, in a few rare situations, there may not be enough blood volume left to assist in resuscitating the newborn. This mechanism as described by Brace needs further research [71].

## 4. Limitations

This article builds on many individual pieces of information obtained through careful human and animal research, which we incorporate to examine how bodily systems work together to accomplish the major life event of neonatal transition. While some of our conclusions are speculative, most are founded on solid physiologic principles and research. We cannot state the ideal time to clamp the cord, as it is not known. There is wide variety as to how long the arterial blood flow continues. We do not yet know if attachment to the low-resistance placenta may have a role in the prevention of neurologic or vascular issues. Optimal practice is likely to wait until the cord is flat and pale or until the placenta is ready to deliver.

## 5. Conclusions

We have presented a case of how intact cord resuscitation with sustained cord circulation supports transition. We have challenged many commonly held beliefs about neonatal resuscitation. The evidence suggests that the current practice of clamping and cutting the umbilical cord and moving the infant away from its mother and placenta is likely an unwise practice. Our case newborn, born depressed and non-vigorous, was fully resuscitated through maintenance of an intact cord. We believe that supporting the process of physiologic neonatal transition may avert potential morbidity or mortality. Placing an emphasis on full perfusion for the neonate after birth may help ensure further advancements in the prevention and treatment of conditions such as ischemic hypoxic encephalopathy and neurodevelopmental injury for infants of all gestational ages.

## Figures and Tables

**Figure 1 children-09-00517-f001:**
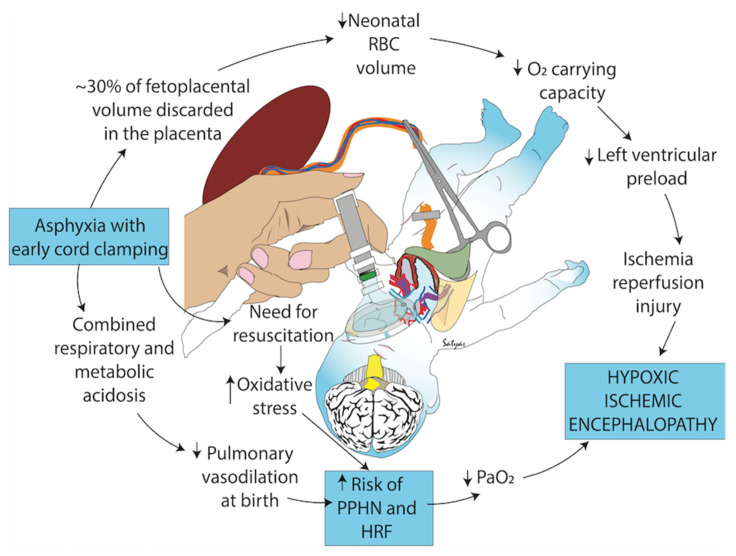
Negative consequences of early cord clamping in depressed infants who may have asphyxia and an increased need for resuscitation. Early cord clamping reduces whole blood and red blood cell (RBC) volume and increases the fetal blood left in the placenta. Hypovolemia and hypoxemia contribute to cerebral hypoperfusion and ischemia and exacerbate pulmonary hypoperfusion and hypertension. Studies have shown increased oxidative stress with early cord clamping compared to delayed cord clamping and umbilical cord milking. Oxidative stress contributes to ischemia and other neonatal morbidities such as hypoxic respiratory failure (HRF), hypoxic-ischemic encephalopathy and persistent pulmonary hypertension of the newborn (PPHN). Copyright Satyan Lakshminrusimha, MD, Sacramento, CA, USA. Used with permission [2].

**Figure 2 children-09-00517-f002:**
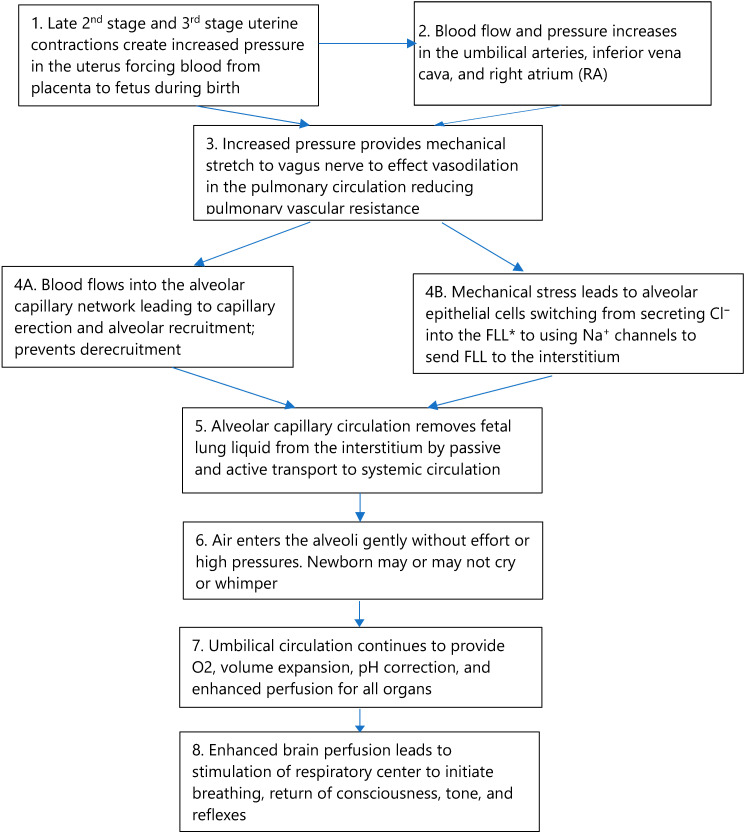
The blood volume model for physiologic neonatal transition.

**Figure 3 children-09-00517-f003:**
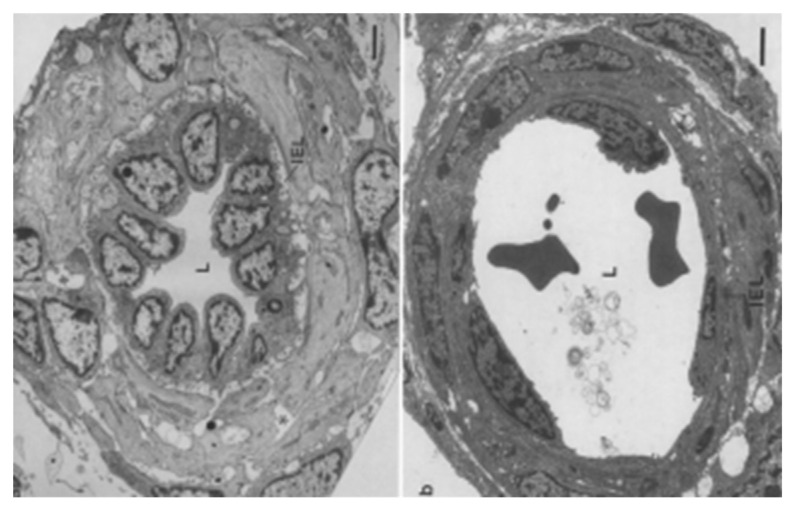
Electron micrographs of transverse sections through small muscular lung arterioles in naturally born piglets in a stillborn on the left and at 5 min of life taken at the same magnification. At birth, the endothelial cells of the intra-acinar arteries showed more rapid and greater changes in shape and thickness than did the cells of more proximal vessels. IEL: internal elastic lamina; L: lumen. Scale bar line on lower right = 2 μm. Adapted from Haworth et al. [27] (with permission).

**Figure 4 children-09-00517-f004:**
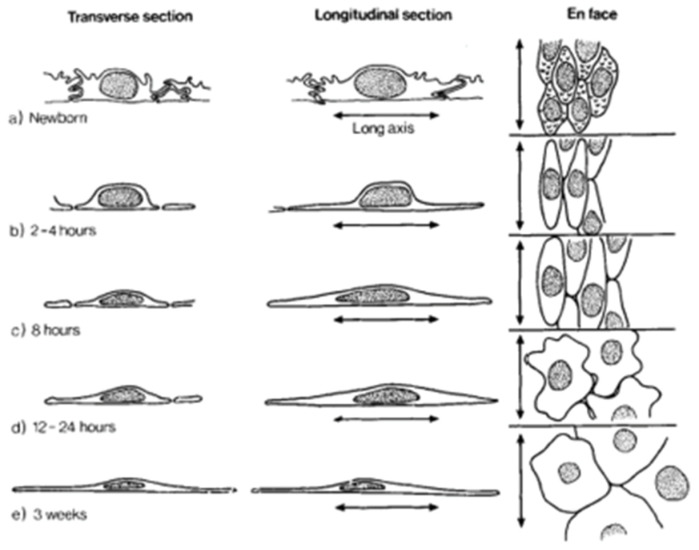
This diagram illustrates shape changes in the endothelial cells of intra-acinar arteries during the first 3 weeks of life (porcine). The endothelial cells of the intra-acinar arteries showed marked changes in cell shape and relationships after birth, while those of large preacinar arteries did not. The first structural changes detected during the first 30 min of life occurred in the endothelial cells lining the intra-acinar arteries. Adapted from Hall and Haworth [31] (with permission).

**Figure 5 children-09-00517-f005:**
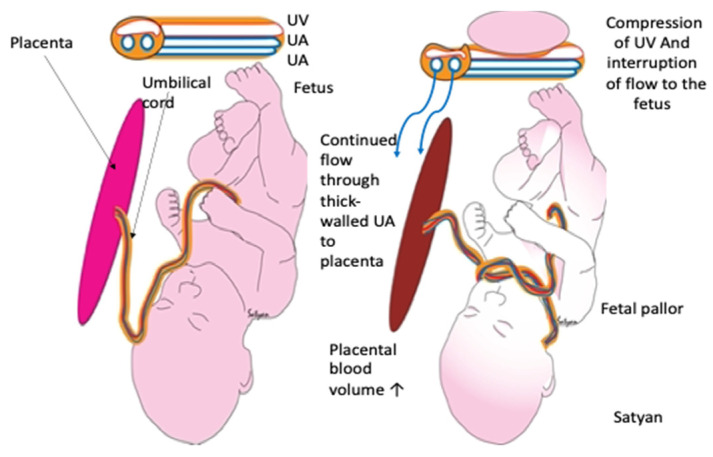
Effects of umbilical cord occlusion on the umbilical vein (UV) and umbilical arteries UA), placenta, and neonate. The “left figure” shows no occlusion; the “right figure” shows compression of the umbilical vein and interference with the flow from placenta to infant. Copyright by Satyan Lakshimrusimha, MD. Sacramento, CA, USA. Used with permission.

**Table 1 children-09-00517-t001:** Composition of Bodily Fluids Compared to Lung Fluid. * (mEq/L). Adapted from Plosa and Guttentag, Lung Development in Avery 10th Edition [20] and Bland R [22].

Component	Lung Fluid	Interstitial Fluid	Plasma	Amniotic Fluid
pH	6.27	7.31	7.34	7.02
Bicarbonate *	3	25	24	19
Protein (g/dL)	0.03	3.27	4.09	0.10
Sodium *	150	147	150	113
Potassium *	6.3	4.8	4.8	7.6
Chloride *	157	107	107	87

**Table 2 children-09-00517-t002:** Some components in cord blood and their role in the body. WBCs, white blood cells; ECs, endothelial cells. (Adapted from Chaudhury 2019; * Disdier 2018).

Factors/Messengers	Role
Angiopoietin	Vascular growth factor
Granulocyte-colony stimulating factor (G-CSF)	Stimulates bone marrow to make granulocytes and stem cells to release them
Bone morphogenic protein-9 (BMP-9)	Regulates iron metabolism; role in memory, learning, attention; bone formation
Endoglin (ENG)	Transmembrane glycoprotein in endothelial cells; growth factor; role in angiogenesis
Endothelian-1 (ET-1)	Peptide with key role in vascular homeostasis; vasoconstriction
Epidermal Growth Factor (EGF)	Transmembrane protein binding; cellular proliferation, differentiation, and survival
Interleukin-8 (IL-8)	Increases angiogenesis and phagocytosis; causes WBCs to migrate to site of injury; chemotaxis
Hepatic Growth Factor (HGF)	Morphogenic factor; paracrine cell growth motility
Heparin Binding EGF-like GF (HBEGF	Glycoprotein; role in heart development and function; would healing
Placental Growth Factor (PGF)	Role in angiogenesis and vasculogenesis
VEGF-A	Acts on ECs, increases vascular permeability, angio- and vasculogenesis, EC cell growth, cell migration; decreases apoptosis
Intra-Alpha inhibitor protein (IAIP) *	Provides anti-inflammatory neuroprotection; reduces production of reactive oxygen species *

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
