# Peer review of "Making the Argument for Intact Cord Resuscitation: A Case Report and Discussion"

_children, 2022, doi:10.3390/children9040517_

Round 1

Reviewer 1 Report

This is an important topic in the context of mother and child health. A few things that may improve the paper are listed below:

  1. Minor editorial input such as line 138 (figure 3[27,24], line 325 can be revised to say 'parasympathetic and sympathetic nervous systems', and line 393 should be '...there may not be enough..'
  2. Table 1 was hard to make sense of.
  3. I wish the case presented included one without the intact cord resuscitation for direct comparison
  4. A more general public-type of question would be how this affects the mother, especially those with chronic conditions such as high blood pressure

Author Response

This is an important topic in the context of mother and child health. A few things that may improve the paper are listed below:

1. Minor editorial input such as line 138 (figure 3[27,24], line 325 can be revised to say 'parasympathetic and sympathetic nervous systems', and line 393 should be '...there may not be enough..'

Thank you. We have taken care of these.

2. Table 1 was hard to make sense of.

Table 1 only compares bodily fluids. Perhaps you were referring to table 2?We agree that Table 2 is complex and have struggled with how to simplify it and still get the information across. It is really just a list of some of the factors found in cord blood and their roles in the human body of which most people remain unaware.

3. I wish the case presented included one without the intact cord resuscitation for direct comparison.

We were limited as to space. However, we have previously published two cases in our Cardiac Asystole paper (doi10.1016/j.mehy.2008.11.019) which discusses some similar points.

 4. A more general public-type of question would be how this affects the mother, especially those with chronic conditions such as high blood pressure.

We appreciate the idea but none of the information to date shows any negative issues affecting the mother. There is no data on women with chronic conditions. A discussion of this would need to be too long to be included. We refer the reader to other recent review articles mentioned in the paper – Rabe, Mercer, Erickson-Owens (https://doi.org/10.1007/s00431-022-04395-x) and Andersson & Mercer (https://doi.org/10.1016/j.clp.2021.05.002)

Reviewer 2 Report

This manuscript is a part of a special volume on Placental/cord management for which there are a team of guest editors

The manuscript is focused on the physiologic changes occurring in the fetus at birth related to a successful transition from intrauterine to extrauterine existence. A case study is presented which has apparently been previously published in a text about Normal birth and the first 2 authors of that chapter are the first 2 authors of this manuscript.  The case report is brief and used for subsequent discussion of the potential mechanisms that contribute to transitional circulation and respiration.

I believe that the paper could be titled "physiologic benefits of delayed cord clamping on neonatal transition"

Infants with nuchal cords will respond favorably to removal of the cord obstruction as did their infant, and the further improvement of the infant may not have been a result of the 20 minutes of continued cord circulation which falls off rapidly after 4-5 minutes. 

The paper is long and detailed and while containing useful information, is not easily read.

The authors make frequent reference to current resuscitation interventions including the use of positive pressure ventilation, with suggestions that such interventions may be harmful. Many of their references are from animal models, and we have learned that such models do not always reflect the human birth process, especially as almost studied animals have received anesthesia and complex instrumentation. If this is to be a balanced presentation a few words about the differences would be appropriate

In addition few of their physiologic hypotheses have been supported by randomized human clinical trials - even a comment that immediate cord clamping was never proven to be an evidence based intervention would be appropriate.

I would suggest the following - as this is a special addition that the guest editors should evaluate the manuscript in light of other contributions and ensure that there is minimal overlap

Shorten the manuscript to be more readable

The Conclusion is very long and should be shortened

Describe the limitations of the arguments made 

There are many references to clinical practice, and as this manuscript focuses on the physiology of transition, I would suggest that these references be removed.

Author Response

Reviewer 1 suggested that the paper could be titled "physiologic benefits of delayed cord clamping on neonatal transition"  

We think that our title gives readers the best idea of what the article covers. One cannot describe a case without describing how one’s actions affected the physiology.

We agree that infants with nuchal cords will respond favorably to removal of the cord obstruction as did our infant.

However, our point is that some of these babies will not respond as well with ICC and may go on to develop serious complications.

We did not mean to imply that the further improvement of the infant may have been a result of all of the 20 minutes of continued cord circulation, but we provided evidence that in many babies the arterial blood flow does last longer than 4-5 min. Plus Polglase’s study showing value of the placenta as a low-resistance reservoir at 10 min in lambs needs further study and consideration. In the absence of more knowledge, we suggest siding with “mother nature” and leaving the cord alone until we have a clear indication that the infant is through with it – when it is ready to deliver.

The paper is long and detailed and while containing useful information, is not easily read.

 We have shortened it considerably, especially the redundancy, and hope it is an easier read.

The authors make frequent reference to current resuscitation interventions including the use of positive pressure ventilation, with suggestions that such interventions may be harmful. Many of their references are from animal models, and we have learned that such models do not always reflect the human birth process, especially as almost studied animals have received anesthesia and complex instrumentation. If this is to be a balanced presentation a few words about the differences would be appropriate.

Agreed. We have been careful to identify whether the research was on human infants or animals. We added “A randomized control trial comparing sustained inflation with usual ventilation in human preterm infants, was stopped because more early deaths occurred in the infants with sustained inflation [52]” to the paper under 3.1.4., line 209. In all fairness, there is only so much experimentation that we can do with human infants. Much of the research done with human newborns is done on those who have already had ICC which changes the physiology considerably.

In addition, few of their physiologic hypotheses have been supported by randomized human clinical trials - even a comment that immediate cord clamping was never proven to be an evidence based intervention would be appropriate.

We added “In many cases, the newborns received immediate or early cord clamping, a non-evidence-based intervention” on page 1 under introduction.

I would suggest the following - as this is a special addition that the guest editors should evaluate the manuscript in light of other contributions and ensure that there is minimal overlap

               Shorten the manuscript to be more readable.

We have shortened it considerably and removed          redundancy. We hope it is more readable.

               The Conclusion is very long and should be shortened.  

We reduced it from four paragraphs to one.

               Describe the limitations of the arguments made.

 We have added a limitations section and addressed    the fact that we may have been quite liberal in our some of the interpretations and associations that we   made. We also added information about the lack of certainty about cord clamping time. See llines 386-395.  

There are many references to clinical practice, and as this manuscript focuses on the physiology of transition, I would suggest that these references be removed.

We removed several of them but since this is a clinical case, and some of them affect the physiology, we have left some where they seen essential.

Round 2

Reviewer 2 Report

The althorns have Madde a Number of charges to this manuscript, and it has been shortened. However most of the original concerns remain. 

The case report is old, and there is a significant concern that removing the unchallenged cord led to the noted improvements and not the prolonged DCC. The authors critisize many resuscitation interventions based on animal studies. While each section is of physiologic interest the relationship to DCC is tenuous.They acknowledge that they may have been quite liberal in their discussion.

My concerns remain - this is an overly lengthy manuscript, most of the discussion is based on animal models with questionable significance to human transition, and the case report is not convincing that DCC was responsible for the infants improvement

Again, I would favor a much shorter, less speculative discussion, and would leave the decision to further modify to the guest editors.
